

# Genome-wide identification and characterization of heat shock protein family 70 provides insight into its divergent functions on immune response and development of *Paralichthys olivaceus*

Kaiqiang Liu[1,2,3], Xiancai Hao[1,2], Qian Wang[1,2], Jilun Hou[4], Xiaofang Lai[3], Zhiguo Dong[3] and Changwei Shao[1,2]

[1] Key Laboratory for Sustainable Utilization of Marine Fisheries Resource, Ministry of Agriculture, Yellow Sea Fisheries Research Institute, Chinese Academy of Fishery Sciences, QingDao, China
[2] Laboratory for Marine Fisheries Science and Food Production Processes, Qingdao National Laboratory for Marine Science and Technology, QingDao, China
[3] Jiangsu Key Laboratory of Marine Bioresources and Environment, Jiangsu Key Laboratory of Marine Biotechnology, Huaihai Institute of Technology, Lianyungang, China
[4] Beidaihe Central Experiment Station, Chinese Academy of Fishery Sciences, Beidaihe, China

Corresponding authors
Zhiguo Dong, dzg7712@163.com
Changwei Shao,
shaochangwei303@163.com,
shaocw@ysfri.ac.cn

## ABSTRACT

Flatfish undergo extreme morphological development and settle to a benthic in the adult stage, and are likely to be more susceptible to environmental stress. Heat shock proteins 70 (*hsp70*) are involved in embryonic development and stress response in metazoan animals. However, the evolutionary history and functions of *hsp70* in flatfish are poorly understood. Here, we identified 15 *hsp70* genes in the genome of Japanese flounder (*Paralichthys olivaceus*), a flatfish endemic to northwestern Pacific Ocean. Gene structure and motifs of the Japanese flounder *hsp70* were conserved, and there were few structure variants compared to other fish species. We constructed a maximum likelihood tree to understand the evolutionary relationship of the *hsp70* genes among surveyed fish. Selection pressure analysis suggested that four genes, *hspa4l*, *hspa9*, *hspa13*, and *hyou1*, showed signs of positive selection. We then extracted transcriptome data on the Japanese flounder with *Edwardsiella tarda* to induce stress, and found that *hspa9*, *hspa12b*, *hspa4l*, *hspa13*, and *hyou1* were highly expressed, likely to protect cells from stress. Interestingly, expression patterns of *hsp70* genes were divergent in different developmental stages of the Japanese flounder. We found that at least one *hsp70* gene was always highly expressed at various stages of embryonic development of the Japanese flounder, thereby indicating that *hsp70* genes were constitutively expressed in the Japanese flounder. Our findings provide basic and useful resources to better understand *hsp70* genes in flatfish.

## INTRODUCTION

Heat shock proteins (HSPs) are a super family of proteins that are induced by physical, chemical and biological stressors in all living organisms from bacteria to humans (*Kregel, 2002*). HSPs were first discovered as genes involved in heat-shock responses in the fruit fly *Drosophila melanogaster* (*Ritossa, 1962*). Based on their roles and expression patterns, HSPs were categorized into two different types: constitutive heat shock proteins (HSCs) that are expressed constitutively, and inducible forms that are expressed in response to certain factors (*Boone & Vijayan, 2002*). HSCs are expressed early in development and are involved in cellular activity, in contrast, inducible HSPs are involved in the response to harmful circumstances and protect the cell from stress (*Angelidis, Lazaridis & Pagoulatos, 1991*; *Whitley, Goldberg & Jordan, 1999*). HSPs have also been classified based on their protein molecular weight, where they are divided into HSP90 (83∼110 KD), HSP70 (66∼78 KD), HSP60 (58∼65 KD) and other small molecular weight proteins (*Morimoto, Tissieres & Georgopoulous, 1990*). Characterization of HSPs in a species genome will facilitate better interpretation of how an organism responds to environmental stressors.

HSP70 are the most conserved HSPs across different species (*Hunt & Morimoto, 1985*; *Mayer & Bukau, 2005*). HSP70 proteins have a characteristic N-terminal ATPase domain, substrate binding domain, and C-terminal domain (*Schlesinger, 1990*; *Kiang & Tsokos, 1998*), the N-terminal ATPase domain, and the substrate binding domain are often more conserved than the C-terminal domain (*Munro & Pelham, 1987*). Humans, birds, amphibians, zebrafish, catfish, and medaka contain 17, 12, 19, 20, 16, and 15 *hsp70* genes, respectively (*Song et al., 2015*). In previous studies, it was shown that *hsp70* genes play fundamental roles as chaperones involved in maintaining cellular function that facilitate protein-folding, regulate kinetic partitioning, and reduce protein aggregation (*Gething & Sambrook, 1992*; *Pratt & Toft, 1997*; *Parsell et al., 1994*; *Morimoto et al., 1997*; *Pratt, 1993*).

HSP70 is a well-known stress protein in aquatic organisms, which is involved in stress response, including thermo tolerance as well as regulating the immune system (*Gornati et al., 2004*; *Poltronieri et al., 2007*; *Bertotto et al., 2011*; *Wallin et al., 2002*; *Tsan & Gao, 2009*). For example, hyper-thermic treatment of *Penaeus monodon* increases *hsp70* expression and reduces the replication of gill associated virus (GAV) (*Vega et al., 2006*). In addition, upregulation of endogenous HSP70 in the *Artemia franciscana* (Kellogg) occurs simultaneously when shielding bacterial infection (*Sung et al., 2009*). Coho salmon infected with *Renibacteriumsal moninarum* expressed higher levels of *hsp70* in the liver and kidney when compared with uninfected salmon, highlighting the importance of *hsp70* genes in immune response of fish (*Forsyth et al., 1997*). Juvenile rainbow trout (*Oncorhynchus mykiss*) infected with *Vibrio anguillarum* has higher *hsp70* expression in hepatic and kidney tissues before showing clinical signs of disease (*Ackerman & Iwama, 2001*). Therefore, *hsp70* is important for the immune response of aquatic species against diverse infections.

In addition to its role in cellular function, stress response and immunity, HSPs have also been shown to be involved in embryonic development and extra-embryonic structures (*Morange et al., 1984*; *Voss et al., 2000*; *Matwee, 2001*; *Louryan et al., 2002*; *Rupik et al., 2006*). During embryonic development, Many HSPs exhibit complex spatial and temporal

expression patterns (*Krone, Lele & Sass, 1997*). For example, mouse embryos treated with anti-HSP70 showed significant reduction in the progression of development (*Neuer et al., 1998*). Zebrafish demonstrated low and constitutive *hsp90a* expression during embryonic development, and these levels increased when the gastrula and later stage embryos were exposed to heat (*Krone & Sass, 1994*). Moreover, *hsp47* showed higher expression in response to stress (*Pearson et al., 1996*), and was involved in the formation of embryonic tissues in fish through its interaction with procollagen (*Krone, Lele & Sass, 1997*). Therefore, HSPs play an important role during embryonic development in addition to their basic cellular functions.

Japanese flounder is endemic to the northwestern Pacific Ocean (*Minami & Tanaka, 1992*). It is the dominant flatfish species in the aquaculture industry because of its rapid growth rate, delicious taste, and high nutritional value, therefore becoming an economically important marine species in China, Korea, and Japan (*Fuji et al., 2006*). The genome of Japanese flounder was recently completed (*Shao et al., 2017*), thereby facilitating the discovery of *hsp70* genes. Here, we identified and characterized the Japanese flounder *hsp70* family and determined whether these genes are involved in stress response to a pathogen, and embryonic development. Comparative genomics between the other closely related species offer a chance to understand the evolutionary relationship of *hsp70* and the selective pressures that affect the evolution of these genes. Our findings provide insight into the function of *hsp70* in embryonic development and disease defense in the Japanese flounder, which may help future improvement of the Japanese flounder for aquaculture.

## MATERIALS & METHODS

### Ethics statement

The handling of experimental fish was approved by the Animal Care and Use Committee of the Chinese Academy of Fishery Sciences, and all protocals were performed in accordance with the guidances of the Animal Care and Use Committee.

### Database mining and sequence extraction

A comprehensive search of the sequence database on the NCBI website and Ensemble website was employed to identify *hsp70* orthologs among six different teleost fish, including: zebrafish, stickleback, medaka, tilapia, platyfish, and tetraodon. Protein sequences of all chosen species were collected, and HSP70 proteins were selected from zebrafish according to the accession number, and HSP70 protein sequences from zebrafish were used as queries to search against the Japanese flounder gene set with an intermediate stringency of e−10. Redundant gene sequences were removed by setting the identity value and coverage of the alignment length to 60% and 60%, respectively. All remaining sequences were manually confirmed for the presence of known HSP70 domains using the software SMART (*Schultz et al., 1998*; *Schultz et al., 2000*) to remove pseudogenes. When applying a similar method, *hsp70* gene sequences were retrieved from the gene set of other species, including stickleback, medaka, platyfish, tilapia, and tetraodon. The Zebrafish Nomenclature Guidelines were used as a benchmark to name *hsp70* genes in flounder.

Furthermore, the isoelectric point (*pI*) of the HSP70 protein was determined using ExPASy (https://www.expasy.org/).

## Phylogenetic analyses

To investigate the phylogenetic relationship of *hsp70* genes among the surveyed fish species, the sequences were processed as follows: protein sequences were aligned using Guidance2 with MAFFT as the MSA algorithm and 100 bootstrap repeats. Ambiguous sites were manually trimmed while aligning sequences. The multiple sequence alignment was used as input into MEGA7 to construct a phylogenetic tree (*Kumar, Stecher & Tamura, 2016*). The phylogenetic relationships of *hsp70* genes of seven teleost fishes were constructed using the ML method in MEGA7. In the ML analyses, the maximum composite likelihood model was used, and a total of 1,000 bootstrap replicates were conducted for each calculation. Finally, Evolview was used to visualize the phylogenetic tree (*Zhang et al., 2012*).

## Sequence structure analysis and motif prediction of *hsp70*

To analyze the gene structure of *hsp70* in the Japanese flounder, the Gene Structure Display Server of Peking University (*Hu et al., 2015*) was used to display the intron and exon structure of all *hsp70* genes. To identify the motif of *hsp70* genes, a structural motif search was conducted using MEME (*Machanick & Bailey, 2011*) with a target motif number setting of 15.

## Molecular evolution analysis

Protein sequences from each clade in the phylogenetic tree were retrieved and used for multiple sequence alignment with Guidance2 (*Sela et al., 2015*). Unreliable sites were trimmed in the multiple sequence alignment, and a tree was constructed using IQ-TREE (*Nguyen et al., 2014*). Codon alignment of protein sequences was converted by pal2nal (*Suyama, Torrents & Bork, 2006*). Using these data, molecular evolution analysis was conducted to measure the selection pressure within each clade, and the CODEML program from PAML (*Yang, 1997*; *Yang, 2007*) was used to estimate the $\omega$ value using the branch site model. The aim of the branch-site test was to identify episodic Darwinian selection along a prespecified branch in a phylogenetic tree that impacts only a few codons in the coding sequence of a gene. Using this model, we detected genes under positive selection and the corresponding sites with a nonsynonymous/synonymous ratio of $\omega > 1$ (*Yang & Nielsen, 2002*; *Yang & Reis, 2011*; *Zhang, Nielsen & Yang, 2005*).

## Structure modeling

To better understand the protein structure of genes under positive selection in Japanese flounder, PHYRE2 (*Kelley & Sternberg, 2009*) was used to predict the protein structure and secondary structure using the default parameter. The sites under positive selection were marked by PyMol 2.0.

## Immune response expression profile of *hsp70* genes against *Edwardsiella tarda* infection in the Japanese flounder

RNA-seq data was downloaded from Sequence Read Archive (SRA) database in NCBI, including the following accession numbers: SRR5713071, SRR5713072, SRR5713073,

SRR5713074, SRR5713075, SRR5713076, SRR5713077, SRR5713078, SRR5713079, and SRR5713080. These data represented the Japanese flounder that was challenged with *E. tar* at 0 h, 8 h, and 48 h, as well as a control injected with Ringer's solution (*Li et al., 2018*). The data was trimmed and the quota transcripts per million of each gene (TPM) was used to display the expression profile of *hsp70* genes.

## Expression pattern of *hsp70* genes during embryonic development of Japanese flounder

The *hsp70* gene expression analysis was conducted during early stages of embryonic development and mature gonads of Japanese flounder. The family of Japanese flounder with crosses of normal females and males were produced and kept in separate units until the collection of samples of sperm, oocytes, the 4 cell stage, 32 cell stage, 128 cell stage, high blastula stage, low blastula stage, early gastrula stage, late gastrula stage, myomere stage, heart beat stage, and hatched larva stage. RNA-seq was conducted on all the above developmental samples (Table S1). In addition, raw sequence data of ovaries and testis were downloaded from NCBI (accession numbers SRR3509719 and SRR3525051). Gene expression levels were assessed using TPM, then the R package pheatmap (*Kolde, 2018*) was used to illustrate the expression patterns at different developmental stages.

## RESULTS

### Identification of *hsp70* superfamily genes

A total of 111 genes were retrieved from seven fish species (Japanese flounder, zebrafish, stickleback, medaka, tilapia, platyfish, and tetraodon), where the number of *hsp70* genes ranged from 9 to 21, depending on the species. There were 9 *hsp70* genes in the tetraodon, whereas tilapia had 21 *hsp70* genes. Fifteen *hsp70* genes, including *hspa1a, hspa4a, hspa12a, hsc70, hspa5, hspa9, hspa1b, hspa12b, hspa14, hspa13, hspa4l, hspa4b, hspa8a, hspa8b,* and *hyou1* were identified in the Japanese flounder (Table 1). All genes contained the necessary domains of *hsp70*. The length of the corresponding protein ranged from 442 to 1,020 amino acids. The *pI* of different genes was variable, ranged from 4.97 to 8.17 (Table 1).

### Phylogenetic analysis of *hsp70* in fish

We next conducted a phylogenetic analysis using 111 *hsp70* genes from seven teleost species (Fig. 1). In our analysis, *hsp70* genes were divided into eight subclades, which matched the known subfamilies of *hsp70* genes. However, we observed ambiguous separation between *hspa1, hsc70,* and *hspa8*. Not all the fish species had genes from each clade. For example, tetraodon did not contain *hspa14* and medaka did not contain *hyou1*. All the members of the flounder *hsp70* were split into distinct clades and were grouped with the corresponding genes from zebrafish and other fish.

### Sequence structure analysis and motif prediction of *hsp70* gene family

In general, *hsp70* genes are variable in length, ranging from from 1839 bp to 21277 bp (Table 1 and Fig. 2). They have diverse numbers of exons, for instance, *hspa1a* and *hspa1b* contained one exon, *hspa4a, hspa4b, and hspa4l* that belong to the same subfamily

**Table 1** Summary of *hsp70* genes in the Japanese flounder genome.

| Name | Accession number | Gene length (bp) | Protein length (aa) | p*I* |
|------|------------------|------------------|---------------------|------|
| *hspa1b* | N_000000250.1 | 1839 | 613 | 5.31 |
| *hspa4a* | N_000000247.1 | 11776 | 834 | 5.13 |
| *hspa12b* | N_000000248.1 | 11247 | 673 | 8.17 |
| *hsc70* | N_000000245.1 | 2998 | 578 | 5.08 |
| *hspa5* | N_000000244.1 | 2767 | 654 | 4.97 |
| *hspa9* | N_000000246.1 | 7683 | 716 | 6.23 |
| *hspa1a* | N_000000243.1 | 1923 | 640 | 5.42 |
| *hspa12a* | N_000000237.1 | 21277 | 655 | 7.3 |
| *hspa14* | N_000000249.1 | 7155 | 506 | 5.96 |
| *hspa13* | N_000000242.1 | 2934 | 442 | 5.5 |
| *hspa4l* | N_000000241.1 | 8479 | 1005 | 5.25 |
| *hspa4b* | N_000000236.1 | 7353 | 835 | 4.98 |
| *hspa8a* | N_000000238.1 | 10619 | 1020 | 6.47 |
| *hspa8b* | N_000000239.1 | 4364 | 659 | 5.32 |
| *hyou1* | N_000000240.1 | 11459 | 970 | 5.12 |

Notes.

p*I* indicates the protein isoelectric point.

contained 19–23 exons. Other genes within the same subfamily shared similar number of introns and exons. The gene structures of *hsp70* from the seven species included in this study are displayed in (Table S2 and Fig. S1). The *hsp70* found in flounder had variable protein motif patterns (Fig. 3). Genes *hspa12a* and *hspa12b* contained three motifs, and *hspa1a* and *hspa1b* contained the maximum number of motifs (15). The motif compositions of different *hsp70* genes are listed in Fig. S2.

## Molecular evolution analysis

Although eight subclades can be found, *hspa1, hsc70,* and *hspa8* clade show ambiguous separation, and could not be used for positive selection analysis. Therefore, we only used data from the other seven *hsp70* subclade genes in Japanese flounder to identify signatures of evolution. We identified four genes, *hspa4l, hspa9, hspa13,* and *hyou1,* as having signatures of positive selection in the Japanese flounder, with $P < 0.05$. Among them, *hspa4l* and *hspa13* contained one positively selected site with posterior probabilities values > 0.95, while *hspa9* contained two positively selected sites. The sites were as follows: the Cys in the protein sequence of gene *hspa4l*, which was the 235th amino acid; the 582th and 587th amino acid Thr were present in the protein of *hspa9*; the His is the 337th amino acid in gene *hspa13* (Table S3).

## Protein structure of genes under positive selection

Next, we generated three-dimensional protein structures of HSPA4L, HSPA9, HSPA13, and HYOU1 using PHYRE2. However, we were unable to predict the structure of HSPA9 and HYOU1. The site under positive selection in significant level was marked in the predicted proteins of HSPA4L and HSPA13 (Fig. 4). The predicted secondary structure of HSPA4L

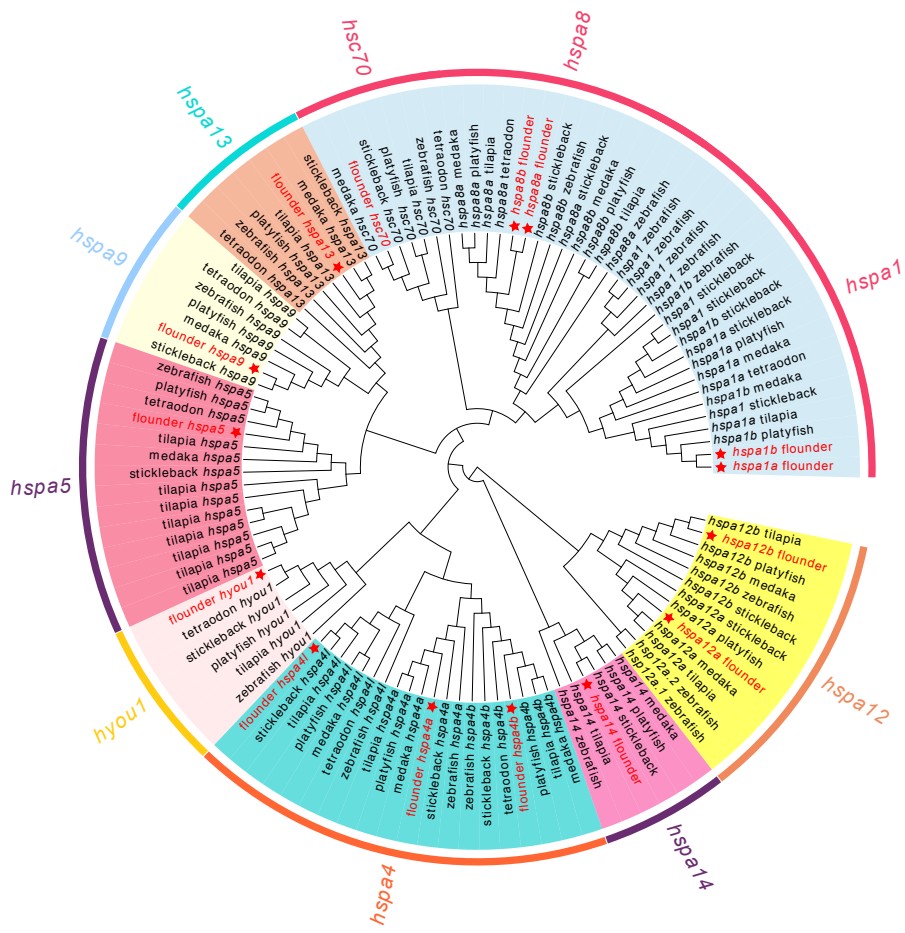

**Figure 1** **Phylogenetic tree of *hsp70* from flounder, medaka, tilapia, zebrafish, platyfish, tetraodon, and stickleback.** The color in the background indicates the branch of sub-family and corresponds to the sub-family names marked in the same color as the circle beyond. The hsp70 genes from flounder are marked with a red star.

demonstrates that the Cys under positive selection is located in a $\alpha$-helix, and the His under positive selection is located in a $\beta$-strand in HSPA13 (Fig. 5).

## Immune response expression profile of *hsp70* genes against *Edwardsiella tarda* infection in Japanese flounder

To test the role of *hsp70* in response to an infection, we analyzed previously generated RNA-seq data of Japanese flounder blood from samples infected with *E. tar*. Overall, the *hsp70* genes showed diverse expression patterns after the *E. tar* infection. Expression levels of *hspa8b, hspa12a, hspa1a, hspa8a, hsc70* and *hspa1b* decreased after 48 h of treatment with *E. tar*. Other genes, such as *hspa9, hspa12b, hspa4l, hspa13,* and *hyou1* showed increased levels of expression after treatment for 48 h. Only the expression of *hspa4a* was similar after 48 h of treatment (Fig. 6). The expression of *hspa1a, hspa4a, hspa9, hspa12b, hspa4l, hspa13,* and *hyou1* was dramatically changed in the samples injected with Ringer's solution

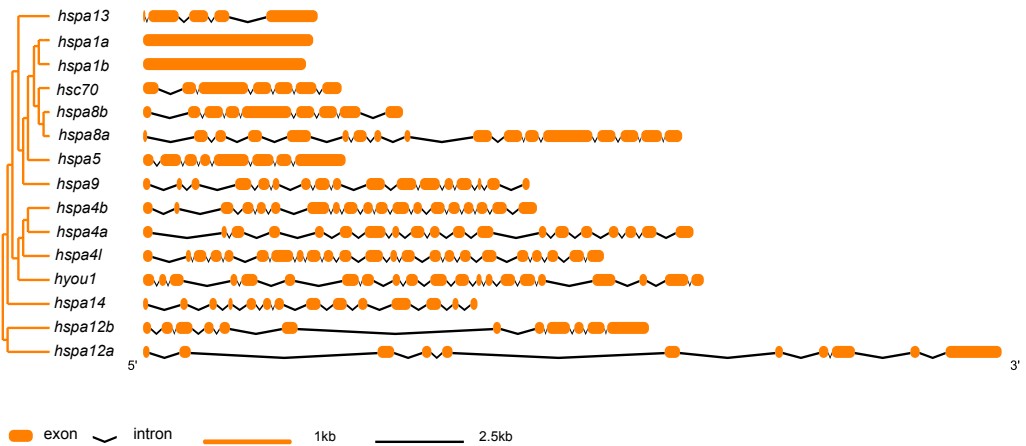

**Figure 2** **Intron-exon structure of *hsp70* genes in flounder.** The phylogenetic tree on the left panel was generated using MEGA7 with the Neighbor-joining (NJ) method and 1,000 bootstrap replicates. The right of the panel shows exon and intron structure of *hsp70*, where the orange rectangles represent exons, black polylines indicate introns, orange and black line indicates scale.

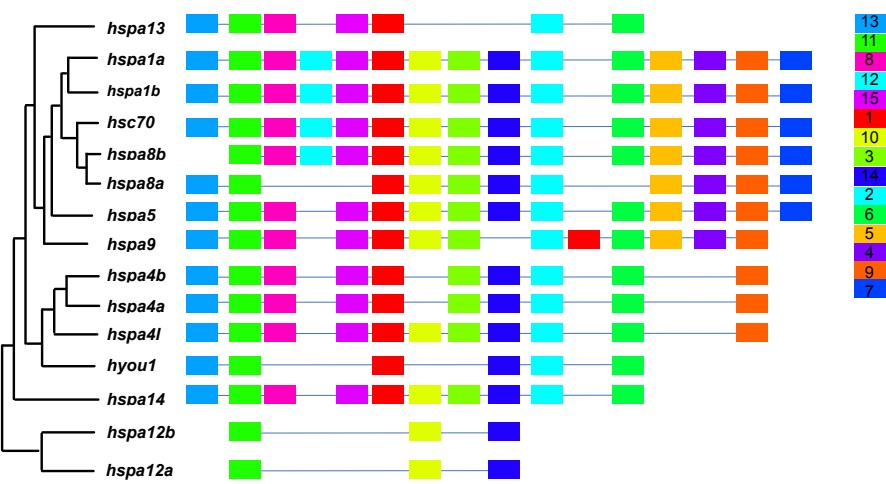

**Figure 3** **Schematic representation of conserved motifs in HSP70 proteins.** Each colored box represents a motif and boxes in the same color indicate the same motif.

after 8 h, however, the expression of genes *hspa12b*, *hspa13* and *hyou1* returned to the original level of expression at 48 h after injection with Ringer's solution.

## Expression pattern in developmental stages of Japanese flounder

We next investigated the expression profile of *hsp70* genes in various developmental stages of the Japanese flounder. We observed significant differences in gene expression based on the developmental stage. Differential expression was observed between oocytes and sperm, where most *hsp70* genes, including *hspa4l*, *hspa4a*, *hspa9*, *hsc70*, and *hspa1b* in oocytes had higher expression level than the sperm. Comparing expression of *hsp70* in sperm and

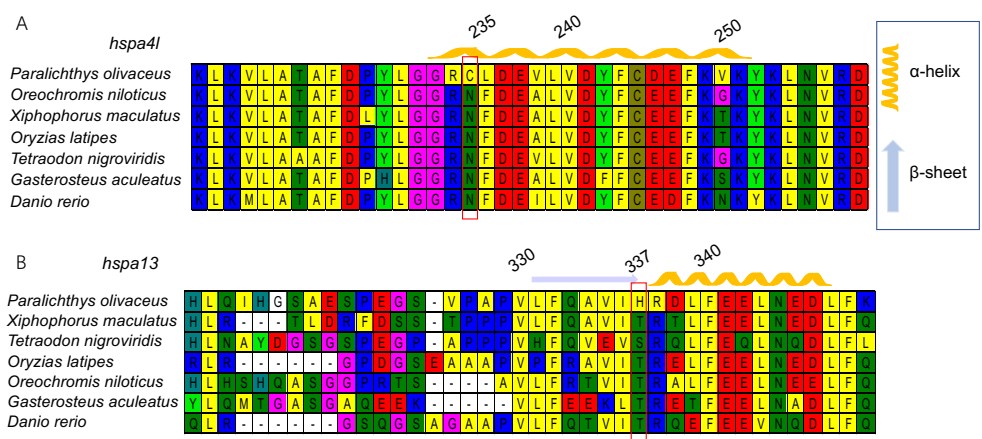

**Figure 4** **Multiple alignments of positively selected sites in *hspa4l* (A) and *hspa13* (B).** The amino acid residue in the red square represents the positively selective site. The secondary structure was predicted by PHYER2, and $\alpha$-helixes were indicated in yellow and $\beta$-sheets were indicated in blue. The number on the top indicates the position of the amino acid residue in the protein.

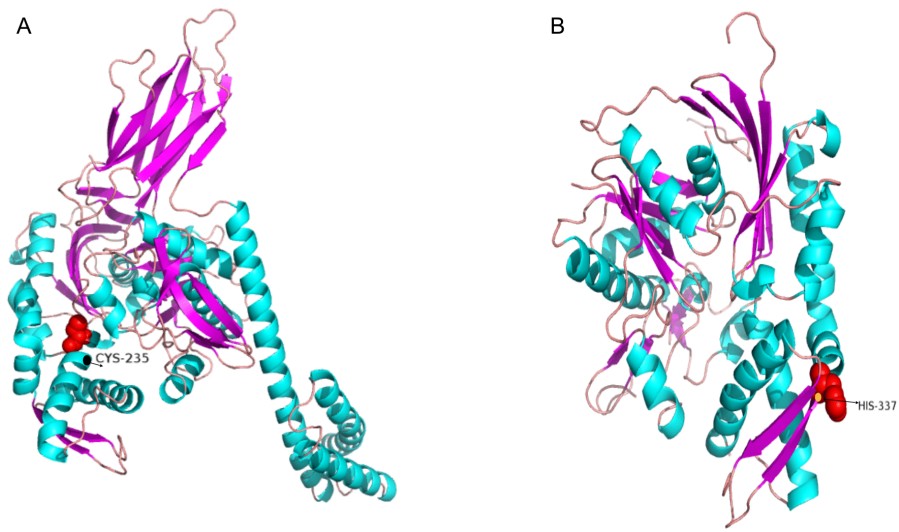

**Figure 5** **The 3D-structural models of HSPA4L (A) and HSPA13 (B).** The amino acid under positive selection in HSPA4L is indicated in black (Cys 235) and located in an $\alpha$-helix. The site under positive selection in HSPA13 is indicated in orange(His 337) and located in a $\beta$-sheet.

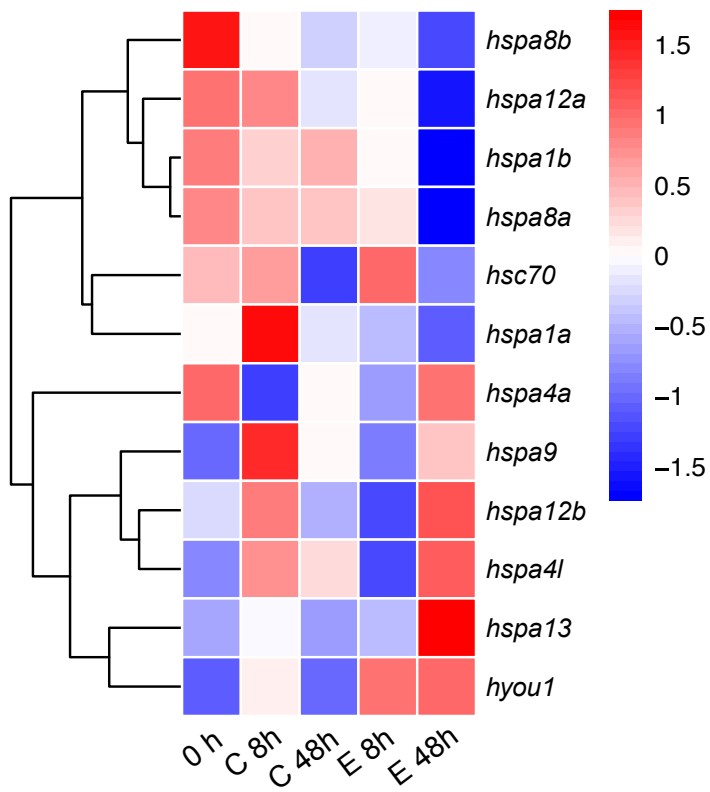

**Figure 6** **Expression patterns *hsp70* in Japanese flounder.** Each column represents a time point, and each row represents a gene. The relative expression level is indicated by the color bar on the top right. 0 h represents the blank control group at the beginning of the experiment, C 8 h, and C 48 h indicates Ringer's solution control group, whereas E 8 h and 48 h indicate a bacteria-challenged experimental group.

testis, some genes, including *hspa4l, hspa4a, hspa9, hspa13, hspa1a* and *hspa8a* had a higher expression level in the testis compared to sperm. When comparing the expression of ovaries and oocytes, some genes, for instance, *hspa1a* and *hspa8a* showed higher expression in the ovaries compared to oocytes, while other genes, for example, *hspa9, hsc70,* and *hspa1b* showed the opposite. In early embryonic development, from oocyte to high blastula stage, *hspa9, hsc70, hspa1b, hspa4l,* and *hspa4a* had high expression. Interestingly, the expression of these genes decreased from the low blastula stage to hatching stage. In contrast, the expression of *hspa8b, hspa13, hspa4b,* and *hspa8a* increased during later developmental stages (Fig. 7).

## DISCUSSION

Studies on HSPs have mainly focused on model organisms such as zebrafish, mouse, and fruit flies (*Rupik et al., 2011*). With increasing genomic data available for other organisms, more in-depth studies can be carried out in a variety of species. Here, we identified and characterized HSPs at the genome level, then explored the evolution of HSPs and its divergent functions on the immune response and different development stages of the Japanese flounder.

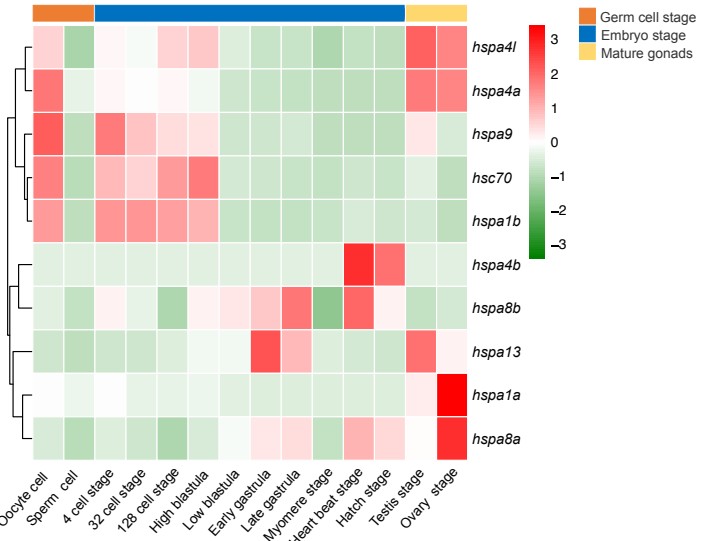

**Figure 7** **Expression profiles of *hspa4l*, *hsp4a*, *hspa9*, *hsc70*, *hspa1a*, *hspa8b*, *hspa13*, *hspa4b*, *hsp8a* and *hspa1b* during the life cycle of the Japanese flounder.** The panel is split into three parts by the three bars on the top, from left to right represents the germ cells, embryonic development stages, and mature gonads. The detailed stages are oocyte, sperm cell, 4 cell stage, 32 cell stage, 128 cell stage, high blastula, low blastula, early gastrula, late gastrula, myomere stage, heart beat stage, hatch stage, testis, and ovary stage. The relative expression level is indicated by the color bar on the top right.

The *hsp70* family genes in Japanese flounder were divided into numbers of branches containing the following genes: *hsc70*, *hspa1*, *hspa4*, *hspa5*, *hspa8*, *hspa9*, *hspa12*, *hspa13*, *hspa14*, and *hyou1*. The phylogenetic relationship and topology of *hsp70* were consistent with previous studies (*Daugaard, Rohde & Jäättelä, 2007*), indicating the confidence of the retrieved sequences in species that were included in the study. Most *hsp70* showed similar intron-exon boundary patterns, suggesting that these genes were highly conserved in fish. However, *hspa8a* (17) had double the number of exons in the flounder compared to other fish (8), and *hspa4l* from all the other species had about 19 exons, whereas the flounder had 23 exons. Interestingly, we found signatures of positive selection in *hspa4l*, further indicating the evolutionary difference of *hspa4l* between flounder and the other species.

New favorable genetic variants sweep population, which is known as positive seletion. (*Wagner, 2007*; *Darwin, 1912*). Genes involved in metabolism, stress response and reproduction tend to be under positive selection (*Oliver et al., 2010*; *Koester, Swanson & Armbrust, 2013*). Among the 15 *hsp70* identified in Japanese flounder, we found signatures of positive selection in four genes, *hspa4l*, *hspa9*, *hspa13*, and *hyou1*, using the branch site model in PAML. Genes under positive selection tended to express less than genes subject to neutral or purifying selection, which tended to be expressed in specific tissues or conditions (*Hodgins et al., 2016*). Purifying and neutral selection tended to affect variants that were deleterious for the organism, and positive selection tended to affect variants that provided an adaptive advantage to the animal (*Rocha, 2006*). Interestingly, *hyou1* was not expressed

at any of the developmental stages. These findings were consistent with previous studies that indicated that genes under positive selection had low expression levels.

The functions of *hsp70* were determined by their cellular location, and intracellular *hsp70* genes protected the cell from stress, while extracellular *hsp70* genes were involved in the immune system (*De Maio, 2014*). For example, *hsp70* could be the cross-presenters of immunogenic peptides in MHC antigens or stimulators that induced innate immune responses (*Pockley, Muthana & Calderwood, 2008*; *Asea et al., 2000*). *Aeromonas hydrophila* challenged with *Labeorohita* showed up-regulation of *apg2*, *hsp90*, *grp78*, *grp75,* and *hsc70*, however, *hsp70* was down-regulated upon infection (*Das, Mohapatra & Sahoo, 2015*). Here, we used RNA-seq data of the Japanese flounder injected with *E. tarda* or Ringer's solution, and we found similar expression patterns as shown in previously published studies (*Li et al., 2018*). However, *hsc70* expression was decreased in Japanese flounder at 48 h after injection with *E. tarda*, which was opposite from the expression pattern of *A. hydrophila*, suggesting a species-specific expression pattern of this gene. Interestingly, some genes were up-regulated shortly after injection with Ringer's solution, and returned to the baseline expression levels after 48 h. However, samples injected with *E. tarda* maintained differences in gene expression even at 48 h after injection. Such divergent expression pattern suggested that some *hsp70* genes were involved in the response to *E. tarda* infection.

Recent studies demonstrated that heat shock proteins play an important role in the sperm–egg recognition and embryonic development (*Li & Winuthayanon, 2017*; *Luft & Dix, 1999*). In mouse, *hsp70* is constitutively expressed from the two-cell to blastocyst stages (*Hahnel et al., 1986*). In this study, from the four-cell stage to the high blastula stage, five genes, including *hspa4l*, *hspa4a*, *hspa9*, *hsc70,* and *hspa1b*, were initially highly expressed, then expression ceased in later stages, besides these five genes also shows highly expression in the oocyte cell. A reasonable conclusion of such a similar expression pattern between the oocyte cell and the early stage of embryonic development is an initial, constitutive burst of *hsp70* expression after boosting the zygotic genome from the four cell stage to the high blastula stage. From the low blastula stage, other genes, for example *hspa8b*, was expressed at a high level, then *hspa13* and *hspa8a,* and *hspa4b* showed highly expresssion in chronological order. Overall, from the beginning of embryonic development to the sexual maturation stage, different *hsp70* genes are highly expressed in various developmental stages. In addition, there is always one or more *hsp70* genes expressed at high-level in the different embryonic development stages. This type of expression during embryonic development has proven that *hsp70* genes were constitutive expression in embryonic development of the Japanese flounder.

## CONCLUSIONS

HSP70 constitutes an important group of proteins that respond to stress. *Hsp70* in the Japanese flounder are divided into eight clades, similar as in other species. Structure analysis of *hsp70* showed that these genes were highly conserved among different species. Four genes were found under positive selection. Genes *hspa9*, *hspa12b*, *hspa4l*, *hspa13,* and *hyou1* were highly expressed in flounders challenged with *E. tarda*, suggesting that these *hsp70* genes

were induced to protect cells from stress. Expression analysis during the developmental stages indicated that *hsp70* genes were involved in embryonic development of the Japanese flounder in a temporal manner. In conclusion, *hsp70* genes play important roles in both the immune response and embryonic development of the Japanese flounder.

### Funding

This study was supported by the National Key R&D Program of China (2018YFD0900301) and the AoShan Talents Cultivation Program Supported by Qingdao National Laboratory for Marine Science and Technology (2017ASTCP-ES06), the Taishan Scholar Project Fund of Shandong of China and the National Ten-Thousands Talents Special Support Program. The International Scientific Partnership Program ISPP at King Saud University for funding this research work through ISPP No. 0050. The funders had no role in study design, data collection and analysis, decision to publish, or preparation of the manuscript.

### Grant Disclosures

The following grant information was disclosed by the authors:
National Key R&D Program of China: 2018YFD0900301.
Qingdao National Laboratory for Marine Science and Technology: 2017ASTCP-ES06.
Taishan Scholar Project Fund of Shandong of China.
National Ten-Thousands Talents Special Support Program.
The International Scientific Partnership Program ISPP at King Saud University for funding this research work through ISPP No. 0050.

### Competing Interests

The authors declare there are no competing interests.

### Author Contributions

- Kaiqiang Liu performed the experiments, analyzed the data, prepared figures and/or tables, authored or reviewed drafts of the paper, approved the final draft.
- Xiancai Hao performed the experiments, approved the final draft.
- Qian Wang and Xiaofang Lai analyzed the data, approved the final draft.
- Jilun Hou performed the experiments, contributed reagents/materials/analysis tools, approved the final draft.
- Zhiguo Dong conceived and designed the experiments, analyzed the data, approved the final draft.
- Changwei Shao conceived and designed the experiments, analyzed the data, contributed reagents/materials/analysis tools, prepared figures and/or tables, authored or reviewed drafts of the paper, approved the final draft.

### Animal Ethics

The following information was supplied relating to ethical approvals (i.e., approving body and any reference numbers):

The handling of experimental fish was approved by the Animal Care and Use Committee of the Chinese Academy of Fishery Sciences, and all protocols were performed in accordance with the guidelines of the Animal Care and Use Committee.

## DNA Deposition

The following information was supplied regarding the deposition of DNA sequences:

The transcriptome data available at CNSA: CNP0000304 (https://db.cngb.org/search/project/CNP0000304/).

The HSP70 gene family sequences are available at CNGB: N_000000236.1, N_000000237.1, N_000000238.1, N_000000239.1, N_000000240.1, N_000000241.1, N_000000242.1, N_000000243.1, N_000000244.1, N_000000245.1, N_000000246.1, N_000000247.1, N_000000248.1, N_000000249.1, N_000000250.1.

## Data Availability

The transcriptome data is available at CNSA: CNP0000304.

## Supplemental Information

Supplemental information for this article can be found online at http://dx.doi.org/10.7717/peerj.7781#supplemental-information.

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

# PeerJ

**Yang Z. 1997.** PAML: a program package for phylogenetic analysis by maximum likelihood. *Bioinformatics* **13**:555–556 DOI 10.1093/bioinformatics/13.5.555.

**Yang Z. 2007.** PAML 4: a program package for phylogenetic analysis by maximum likelihood. *Molecular Biology and Evolution* **24**:1586–1591 DOI 10.1093/molbev/msm088.

**Yang Z, Nielsen R. 2002.** Codon substitution models for detecting adaptation at individual sites along specific lineages. *Molecular Biology and Evolution* **19**:908–917 DOI 10.1093/oxfordjournals.molbev.a004148.

**Yang Z, Reis MD. 2011.** Statistical properties of the branch-site test of positive selection. *Molecular Biology and Evolution* **28**:1217–1228 DOI 10.1093/molbev/msq303.

**Zhang H, Gao S, Lercher MJ, Hu S, Chen WH. 2012.** EvolView, an online tool for visualizing, annotating and managing phylogenetic trees. *Nucleic Acids Research* **40**:569–572 DOI 10.1093/nar/gks576.

**Zhang J, Nielsen R, Yang Z. 2005.** Evaluation of an improved branch-site likelihood method for detecting positive selection at the molecular level. *Molecular Biology and Evolution* **22**:2472–2479 DOI 10.1093/molbev/msi237.