# Peer review of "Genome-wide identification and characterization of heat shock protein family 70 provides insight into its divergent functions on immune response and development of Paralichthys olivaceus"

_PeerJ, doi:10.7717/peerj.7781_

## Round 0.1 · original submission · Major Revisions

I have heard back from three reviewers. Two reviewers are quite positive, and have suggested some constructive changes to your work, while the third reviewer was much more critical of your work, pointing out the lack of model selection, and more importantly, the lack of a clear hypothesis or goal for your work. Additionally, the two more positive reviewers (and myself) note that the English needs much work.

I hesitated between reject and major revision, but have decided to go with major revision as I believe the reviewers' scientific comments can be addressed by you and your co-authors. However, along with the scientific comments, I request you to have your paper professionally edited, and to provide me with the certificate or name of who edited your work. I have noticed recently some such services actually do not do a very good job, so please choose someone or some service with a good reputation. As PeerJ does not provide editing services, bringing the English up to international standards is the job of the authors and not editors or reviewers.

I look forward to seeing a revised version of your work.

New:

Reviewer 1 ·

Basic reporting

It is an interesting study that provided some insight into hsp70 function on immune response and early development of an commercial important (in aquaculture) flounder species (Paralichthys olivaceus). In other words, they: i) identified and characterized HSPs at the genome level, ii) explored the evolution of HSPs and iii) its divergent functions on immune response and development stage. As major findings, and summing up: a) four genes were found under positive selection, b) hspa9, hspa12, hspa4l, hspa13 and hyou1
were induced to protect cells from stress, and c) analysis during the developmental stages indicated that hsp70s are indeed involved in the flatfish embryonic development. The paper (and respective sections) is well-structured, the methods are sound, and the conclusions supported by the obtained results. The number of Figures (#6) and References are adequate, as well as the length (number of words) of the entire ms.
The paper should be accepted after minor revisions.


Specific comments

The English language should be greatly improved.
E.g1. first line of the abstract “… and settle to a benthic in adults…”. This sentence does not make sense.
E.g.2. “…early in organismal development and are involved in cellular.”. This sentence does not make sense.
Many other examples can be found throughout the manuscript.

Experimental design

Clear. well defined.

Validity of the findings

They are statistically sound and the data quite robust.

Additional comments

NA

Reviewer 2 ·

Basic reporting

The manuscript titled “Genome-wide identification and characterization of
heat shock protein family 70 provides insight into its divergent functions on immune response and development of Paralichthys olivaceus” by Liu et al. reports the identification of 16 hsp70 genes in the genome of flatfish Paralichthys olivaceus. Characterization of these heat shock protein hsp70 s provided insights into the function of these proteins in embryonic development and in immune response of Paralichthys olivaceus. The manuscript is overall well-structured and focused. The findings are interesting and will be utilized by the scientific community. However, I have found many grammatical errors, spelling mistakes and ambiguous sentences that need to be corrected.
Authors should also discuss the findings of Qi J. et al 2014 (doi: 10.1016/j.fsi.2014.05.038) and Yong-hua Hu et al 2012 (doi.org/10.1016/j.fsi.2012.07.015) in this manuscript.

Experimental design

The experiments are well executed.

Validity of the findings

Findings are certainly interesting and will be helpful.

Additional comments

My major concerns are listed below:

Line 58: Please complete the sentence “HSC are expressed early in organismal development and are involved in cellular”.

Line 130: Replace “calculated” with “determined”.

Line 170: Lines 170-171: Please revise the subheading for the following section.

Line 204: Please provide the protein sequence alignment figure to support that hspa1b and hspa1c proteins are highly conserved.

Line 221: What does authors mean by this “hspa1a and hspa1b contained most motifs”? Please write the sentence clearly.

Line 227: Please correct “Among of them”. Remove “of”.

Line 228-229: Please rephrase the sentence “The sites were as follows: a Cys in 235 aa in hspa4l, His in 562 aa in hspa9 and His in 327aa in hspa13”, to make it clear and understandable by the future readers of the manuscript. Please take help of the other published papers for the style others opt for reporting similar results.

Line 240-241: Please revise the subheading for the following section.

Line 309: Please provide references.

Line312: Correct “thatinduce”

Line 317: “48h”, leave space between unit and time, do same wherever applicable.
Line 318 and 322: Replace “A.hydraphila with A. hydraphila” and E.tarda with E. tarda.


Manuscript is well structured, focused and experimentally sound. However, there are grammatical and spelling errors in the writing of the manuscript, Authors need to do the proofreading to avoid such errors. Manuscript also need to be revised for the quality of language used.

Reviewer 3 ·

Basic reporting

In this manuscript, all the data were not verified by experiment but only data analysis. The expression of E.tar challenge and embryonic development was only identified by RNA-seq. Further, the Raw data was downloaded from NCBI (Li et al., 2018, Dev. Comp. Immunol.), and no qRT-PCRs were performed. The whole manuscript lacks novel opinion. The objective of the manuscript was ambiguous, evolution? Immunology? sex differentiation or embryonic development?

In the positive selection analysis, the authors ignored model selection. The best model should be selected or calculated before constructing phylogenetic tree, such as jModelTest or ProtTest. In this study, only six different species were used in positive selection. The number of teleost was not enough for PAML (Yang, 1997). The results of PAML about M0 vs M3, M1 vs M2 and M7 vs M8 (a) were not described.

The authors did not express clearly about the purpose of expression analysis during the developmental stages using DH & ND. The discussion about hsp70s expression in two half-sibling families is not convincing. The high expression level of hsp70 may cause low surviving rate? Further, Fig7 is not clearly arranged and easy to understand.

Experimental design

no comment

Validity of the findings

no comment

---

## Round 0.2 · Minor Revisions

I have heard back from two reviewers, who are satisfied with your work. From a scientific point of view, I agree with them. However, in the previous decision, I noted:

...along with the scientific comments, I request you to have your paper professionally edited, and to provide me with the certificate or name of who edited your work. I have noticed recently some such services actually do not do a very good job, so please choose someone or some service with a good reputation. As PeerJ does not provide editing services, bringing the English up to international standards is the job of the authors and not editors or reviewers.

You did not provide concrete details on who or what company performed editing, and the manuscript as it stands now still has many English problems - more than a simple small amount of editing can deal with at the proof stage. Thus, I repeat my request for you to have your paper professionally edited, and to provide me with the certificate or name of who edited your work. Unfortunately, I cannot accept your paper until it reaches international standards, and hope you can understand my position on this matter.

I look forward to seeing your newly revised version.

[]

Reviewer 2 ·

Basic reporting

I am satisfied with the revision. Authors answered all my concerns and improved the manuscript wherever required.

Experimental design

The design of experiments is well plotted and well executed by the authors.

Validity of the findings

The findings are interesting and will be utilized by the scientific community. However most of the conclusions are presented based on the bioinformatic evidences only.

Additional comments

Manuscript is well structured and focused and can be considered for publication in PeerJ

Reviewer 3 ·

Basic reporting

I think this revised MS should be accepted.

Experimental design

The experiments are well designed and executed.

Validity of the findings

Findings are interesting and helpful.

External reviews were received for this submission. These reviews were used by the Editor when they made their decision, and can be downloaded below.

---

## Round 0.3 · Minor Revisions

Unfortunately, while the English is improved, there are still many mistakes. I have noted twice:

"...along with the scientific comments, I request you to have your paper professionally edited, and to provide me with the certificate or name of who edited your work. I have noticed recently some such services actually do not do a very good job, so please choose someone or some service with a good reputation. As PeerJ does not provide editing services, bringing the English up to international standards is the job of the authors and not editors or reviewers.
You did not provide concrete details on who or what company performed editing, and the manuscript as it stands now still has many English problems - more than a simple small amount of editing can deal with at the proof stage. Thus, I repeat my request for you to have your paper professionally edited, and to provide me with the certificate or name of who edited your work. Unfortunately, I cannot accept your paper until it reaches international standards, and hope you can understand my position on this matter."

You did not provide me with a name or certificate of who edited your paper, and there are still mistakes. PLEASE follow my instructions above, or else I will have to reject this paper upon resubmission without such information and corrected English. While I can appreciate English is difficult, there are international standards to be upheld, and your failure to follow my advice is delaying acceptance of your work.

External reviews were received for this submission. These reviews were used by the Editor when they made their decision, and can be downloaded below.

---

## Round 0.4 · accepted · Accept

Thank you for your diligence on this submission; I am happy to move this into production.

External reviews were received for this submission. These reviews were used by the Editor when they made their decision, and can be downloaded below.